# Mobile In-Ear Power Sensor for Jaw Joint Activity

**DOI:** 10.3390/mi11121047

**Published:** 2020-11-27

**Authors:** Jacob Bouchard-Roy, Aidin Delnavaz, Jérémie Voix

**Affiliations:** Mechanical Engineering Departement at École de Technologie Supérieure, Montréal, QC H3C 1K3, Canada; jacob.bouchard-roy.1@ens.etsmtl.ca (J.B.-R.); aidin.delnavaz@ens.etsmtl.ca (A.D.)

**Keywords:** jaw joint activity, in-ear power sensor, earcanal dynamic movement, energy harvesting

## Abstract

In only a short time, in-ear wearables have gone from hearing aids to a host of electronic devices such as wireless earbuds and digital earplugs. To operate, these devices rely exclusively on batteries, which are not only cumbersome but known for several drawbacks. In this paper, the earcanal dynamic movements generated by jaw activity are evaluated as an alternative source of energy that could replace batteries. A mobile in-ear power sensor device capable of measuring jaw activity metrics is prototyped and tested on three test subjects. The test results are subsequently analyzed using a detection algorithm to detect the jaw activity based on the captured audio signals and to classify them into four main categories, namely chewing, swallowing, coughing and talking. The mean power associated with each category of activity is then calculated by using the pressure signals as measured by a water-inflated earplug subjected to earcanal dynamic movement. The results show that 3.8 mW of power, achieved mainly by the chewing movement, is readily available on average from within the earcanal.

## 1. Introduction

Battery technology appears to be the slowest evolving feature in hearing aids. While hearing aids are becoming more compact with better sound quality, and enhanced features, their power supply battery continues to occupy a high ratio of their total volume. Most hearing aids rely on electrochemical batteries to operate, which involves either replacing or recharging them periodically, resulting in limited energy autonomy and reduced comfort to users. A possibility to overcome these disadvantages is to harvest energy from the human body or the proximate environment.

Several types of hearing aids whose chargers are powered by alternative energy sources have already been developed. A methanol-based micro fuel cell has been used for the instant charging of hearing aid batteries [1]. Moreover, solar-powered chargers help to make hearing aids more affordable to the consumer and less pollutant to the environment [2]. However, these technologies have done nothing to improve the energy autonomy of hearing aids.

This is why alternative sources of energy must be found to replace batteries. The energy required to power hearing aids can be extracted from electromagnetic waves. It is estimated that a person who spends 1 h outdoors every day will be able to harvest an average power of 500 μW from ambient light and 4 μW from radio waves [3] based on the most optimistic scenarios.

Body heat is another source of alternative energy for hearing aids. The human head could provide 0.60–0.96 W of power in an optimal energy conversion situation [4]. A thermoelectric generator and its associated electronic circuits have been tested in the area around the ear [5], but the practicality, comfort and efficacy of such systems are relatively limited given that an array of generators and consequently more skin surface are eventually required to obtain the necessary amount of power.

Ear-centered energy sources for hearing aids are more interesting for obvious reasons. Take for instance the Endocochlear Potential (EP) that is created due to a difference in the number of potassium ions (charged atoms) on either side of a membrane deep in the inner ear [6]. Findings show that this kind of biological battery could harvest around 1 nW from the cochlea of a guinea pig for up to 5 h. Another ear-centered energy source is the dynamic movement of the earcanal. The jawbone is connected to the skull by the Temporomandibular Joint (TMJ), which is anatomically located very close to the earcanal near its first bend. Consequently, jaw movements deform the earcanal, causing what is called earcanal dynamic movements.

Earcanal dynamic movements have been studied for several reasons by researchers. In particular, audiologists are seeking to enhance the comfort of hearing aids and solve their associated retention problems [7,8,9]. Indeed, the magnitude of the canal’s widening should be correctly quantified by clinicians during patient examination based on the earmolds taken at two extreme jaw positions: wide open and closed, to correctly select an impression material and decide whether or not taking an open mouth impression is justified [10]. Moreover, earcanal dynamic movements have been investigated for sensory applications. An earcanal bending sensor consisting of a thin piezoelectric strip attached to a custom-fitted earpiece has been developed to estimate the average bending moment and the resulting stress applied to a custom-fitted earpiece while opening the mouth [11]. Furthermore, a set of infrared proximity sensors has been utilized to measure the earcanal dynamic movements for different applications that require jaw activity tracking, such as silent speech recognition, jaw gesture detection and food intake monitoring [12].

The idea of energy harvesting from earcanal dynamic movements was first presented by the authors in 2013 [13]. Two different mechanisms (electromagnetic and piezoelectric) were proposed to harvest energy associated with such movements. The prototypes were tested for a single test subject. Based on the results, the energy capability of the earcanal was subsequently roughly evaluated for 12 test subjects based on the extent of their earcanal deformations [14]. The earcanal dynamic movements were later investigated more extensively by either COMSOL (Stockholm, Sweden) multiphysics finite element modeling [15] or analytical modeling [16] to evaluate the energy capability of the earcanal dynamic movements. A more recent study on a wearable power sensing device developed to measure the power associated with earcanal dynamic movements [17] shows that a mean power of 26.2 mW is readily available on average for a single earcanal, based on a test in which 6 subjects chewed a meal for a period of several minutes. However, the paper does not provide information about the energy capability of the earcanal dynamic movements for other jaw joint activity nor on the long-term energy evaluation tests.

This paper aims to calculate the energy capability of the main jaw joint activities, namely mastication, swallowing, coughing and talking during a 3-h earcanal dynamic evaluation test for three test subjects. The in-ear power sensor equipped with an earphone to detect jaw joint activity is presented in the next section. Section 3 and Section 4 discuss details to calculate the power and detect and classify the jaw joint activity using the developed in-ear power sensor. Tests and measurements are presented in Section 5 followed by results and discussions in Section 6. Finally, the conclusions are drawn in Section 7.

## 2. In-Ear Power Sensor

The mobile in-ear power sensor device is composed of a pair of earmuffs; that is, a headband that fits over the head supporting two instrumented cups at each end. The right cup consists of a pre-installed inflatable earplug with three levels of earcanal penetration: short, medium and long. The earplug is inflated with water using a hydraulic circuit including tubes, valves and a pressure sensor, all fitted inside the right cup as shown in Figure 1. This cup also contains electronic circuits for pressure signal conditioning, amplification and data transfer. Water is injected into the hydraulic circuit using a syringe, through the input check valve and the signals are transmitted to the data acquisition and recording system through wires coming out of the cup.

The left cup is equipped with an earphone with two miniature microphones: In-Ear Microphone (IEM) and Outer Ear Microphone (OEM) as shown in Figure 2. The earphone is fitted into the earcanal using a generic foam earpiece called EARTIP (Comply, Oakdale, MN, USA).

The Auditory Research Platform (ARP 3.0) and its associated data acquisition system developed in the authors’ laboratory (CRITIAS, Montréal, QC, Canada) are used for both audio and pressure signal acquisition. The ARP uses a tactile screen and Windows operating system and is placed inside a waist bag carried by the subject during the test as shown in Figure 3. A rechargeable battery is also placed in the bag to power the pressure sensor and the analog front-end by a wire running to the earcup.

## 3. Jaw Joint Power Calculations

The total energy of the liquid-filled earplug inside the earcanal can be calculated by the following energy equation
(1)$$E=E_{i}+E_{e}=PV+\frac{1}{2}kU^{2}$$
in which the first term is the internal energy ($E_{i}$) of the water at pressure *P* and total volume *V*, and the second term is the elastic potential energy stored in the expandable earplug membrane and the deformable earcanal wall with the equivalent stiffness $k=\frac{P}{U}$ and the total deformation *U* measured with respect to a reference position $U_{0}$. Upon jaw joint activity, the kinetic energy of the earcanal movement is transferred to the internal pressure energy of the water and the elastic deformation of the earcanal–earplug system. By assuming that the water is incompressible (dV=0) and ignoring the energy loss due to dry friction, viscous damping or heat dissipation, the instantaneous power can be calculated by the time derivation of the energy equation, whereby
(2)$$W=\frac{dE}{dt}=\frac{VdP}{dt}+\frac{PdP}{kdt}$$

The pressure variation d*P* is measured by the pressure sensor and the equivalent stiffness *k* of the combined earplug and earcanal system is experimentally measured for each test subject by measuring the rate of the pressure change (measured by the pressure sensor) with respect to the volume change of the earplug (measured by the syringe with precise volume markings) as the earplug inside the subject’s earcanal is being filled. Therefore,
(3)$$W=\frac{dE}{dt}=W_{i}+W_{e}=\frac{VdP}{dt}+\frac{PdP}{kdt}$$

The total energy *E* during the time interval Δt is calculated by integrating (Equation 3) over the time period, as
(4)$$E=\int_{0}^{\Delta t}Wdt$$

Finally, the average power generated by the earcanal’s dynamic movements during the test $W_{\mathrm{mean}}$ is estimated by dividing the total energy from Equation (Equation 4) by the duration of the test, Δt, shown as
(5)$$W_{\mathrm{mean}}=\frac{E}{\Delta t}$$

## 4. Jaw Joint Activity Detection

The verbal and nonverbal signals captured inside and outside of the earcanal can be used to detect and classify the different types of jaw joint activity. By occluding the earcanal using the eartip, the external sounds are attenuated while the internal sounds including those generated by jaw joint activity are amplified. Both external and internal audio signals are recorded and used by the detection algorithm that was recently developed based on the Gaussian Mixture Model (GMM) to detect verbal and nonverbal human-produced audio events as listed in Table 1 [18]. The highest accuracy achieved by this algorithm for the detection of nonverbal events is 75%.

The events listed in Table 1 are not all associated with jaw joint movements. Those that are have been grouped in the second part of Table 1. Mastication, swallowing, coughing and talking are four main types of mandibular activity. Chewing or mastication is the process by which food is physically crunched and ground by the teeth and chemically broken down by saliva. After chewing, the food is swallowed. Therefore, the clicking of teeth and saliva noise are both considered an indicator of mastication. Sometimes swallowing is not preceded by mastication, mainly for activities such as drinking or sipping. Therefore, saliva noise when it happens alone can be related to swallowing. Coughing is characterized as a sudden expulsion of air through the large breathing passages that normally involves jaw movement and can be detected directly by the developed algorithm. Finally, talking can be easily detected by the speech processing function integrated within the algorithm.

## 5. Tests and Measurements

The human subject test procedure was approved by the “Comité d’éthique de la recherche”, École de technologie supérieure’s internal review board. Three male subjects having no malformation of the earcanal and being in good health participated in the test. Each test subject was asked to put on the earmuffs with the earphone and inflatable earplug in the left and right ears, respectively, and attach the ARP bag around his waist. Then, using a syringe, the hydraulic circuit was filled and flushed with water to remove any air bubbles. The connections were checked for leaks. While the earplug was being filled with water, the pressure signal was recorded at every 0.05 mL of water injected to calculate the equivalent stiffness until it reached the final pressure of 14 kPa, which has been established as the required pressure for a good fit [19]. Once the fit was obtained, the syringe was removed and the measurements began. Since the test setup is fully portable and mobile, the test subject was permitted to go on with his normal daily activity at work while the test went on. The tests started around 10:30 a.m. and lasted about three hours to include lunchtime. During the test, the pressure data were recorded in a tabulated text file (.txt) while the audio signal was recorded in a PCM (Pulse-Code Modulation) waveform audio format (.wav). Both signals were recorded using the ARP’s clock to ensure the time synchronization of the signals. At the end of the test, the water in the hydraulic circuit was collected, measured and used for subsequent calculations.

## 6. Results and Discussion

The pressure–volume diagram of the earplug as it is being filled while inside the test subject’s earcanal is illustrated in Figure 4. This graph can be used to determine the rigidity *k* of the earplug–earcanal system by calculating the tangent of the curve at the working pressure of 14 kPa.

The audio and pressure signals measured for one test subject are shown in Figure 5. This figure also includes the instantaneous power computed using Equation (Equation 3). Two dense areas of pressure variations can be identified in Figure 5b which result in two high power regions around 4000 s and 9000 s in diagram (c). Correspondingly, the audio activity can be evaluated as high around 4000 s and low around 9000 s as depicted in diagram (a). Therefore, it is quite reasonable to assume that the first high mandibular activity region is associated with continuous talking while the second is related to eating which is normally more quiet than talking. This assumption can be validated by listening to the audio signal. However, given the long duration of the tests and the number of test subjects, a listening-based interpretation of the audio signals would be very time consuming.

Therefore, the results are fully investigated by programming the jaw activity detection algorithm based on the audio signals. For each activity detected by the algorithm, an identifier is given to the time axis to mark the time interval at which the detected verbal and non-verbal events happen. Then the corresponding power signal is segmented based on the defined identifiers and the segments are regrouped into five categories: (1) chewing (2) swallowing, (3) coughing, (4) talking and (5) no activity. The detailed results for one test subject are shown in Figure 6.

According to Figure 6, the variation of the calculated power associated with chewing is quite consistent given the regular rhythm of chewing that a person normally maintains while eating. The power associated with talking is more volatile depending on the uttered words, speaking pace and voice loudness. Swallowing exhibits more discrete power regions due to its low frequency profile. Finally, the power diagram for coughing shows more distinct and isolated peaks due to its spontaneous occurrence.

A high instantaneous power peak does not necessarily mean that the corresponding jaw activity has the greatest potential for energy harvesting. For example the coughing event seen in Figure 6 can generate more than 300 μW at its maximum peak, which is much higher than the maximum power peak for the chewing event. However, coughing happens sporadically and lasts less than a second, while chewing is continuous and lasts several minutes. Therefore, the segmented power diagrams are subsequently used in Equation (Equation 4) to obtain the total energy and eventually Equation (Equation 5) is used to calculate the mean power associated with each jaw joint activity. The mean power generated by the internal pressure ($W_{i}$) and elastic deformation ($W_{e}$) components as well as the duration of each jaw movement-related event are reported in Figure 7 for all test subjects.

As shown in Figure 7, chewing has the longest lasting sequence among the test subjects with the duration varying from 5 to 21 min. Talking comes in second place with a duration of less than 8 min for all test subjects. Coughing comes in last, as it only lasts a couple of seconds. The classification of the mean power among test subjects is quite varied. While chewing has the greatest mean value for subject 1, swallowing and coughing generate more power on average for subjects 2 and 3. Nonetheless, the grand winner of the maximum energy potential (the multiplication of the mean power and duration) is chewing (except for subject 2, for whom talking generates as much energy as chewing).

A summary of the results, parameters and variables used in the paper to calculate the energy potential and power generation for each test subject is listed in Table 2. This table presents the water volume *V*, and the equivalent rigidity *k* taken into consideration in Equation (Equation 3) to calculate the instantaneous power. In addition, the total duration of the test δt, the total energy potential *E* and the calculated mean power $W_{\mathrm{mean}}$ are presented for each test subject separately in this table. The table also includes *R*, which represents the time ratio at which the jaw joint is active. The mean value of *R* is 13%, which means that the jaw bone was stationary and inactive for most of the time. Gum chewers and talkative people are likely to have higher jaw activity time ratios and hence more potential to generate in-ear energy. In addition, according to the results, 3.8 mW of power on average are expected to be available from earcanal dynamic movements, which is about 4 times greater than the power needed to run typical hearing aids.

## 7. Conclusions

Earcanal dynamic movement generated by the main jaw joint activities was investigated in this paper and found to be a promising source of energy to power hearing devices. The portable sensor device developed in this paper could successfully measure the audio and pressure signals inside the earcanal, and transmit and store the information in a mobile computer platform. The jaw joint activity detection algorithm could efficiently detect and classify the jaw activities. In addition, the analytical model of the inflated earplug inside the earcanal could estimate the available power in the form of internal pressure and elastic deformation separately. Finally, the energy capacity associated with each jaw joint activity was quantitatively evaluated and reported for three test subjects. The results show that 3.8 mW of power on average is generated in one test subject’s earcanal with chewing having the greatest energy source potential among all types of jaw activities. The conversion from kinetic energy to electrical energy can be accomplished among others via piezoelectric effects in the future. However, other human factors like ergonomics, comfort and metabolic cost of such in-ear energy harvesting devices should be thoroughly investigated in advance.

## Figures and Tables

**Figure 1 micromachines-11-01047-f001:**
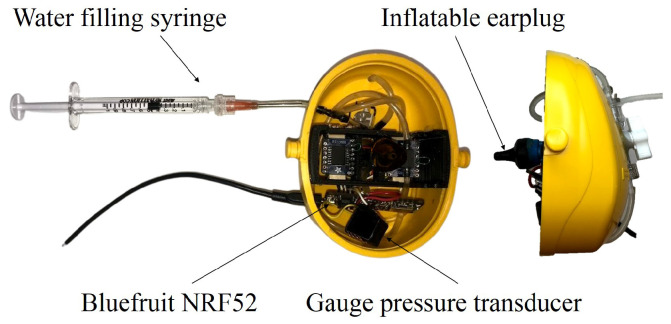
In-ear power sensor: inflatable earplug and associated hydraulic circuit in the right cup.

**Figure 2 micromachines-11-01047-f002:**
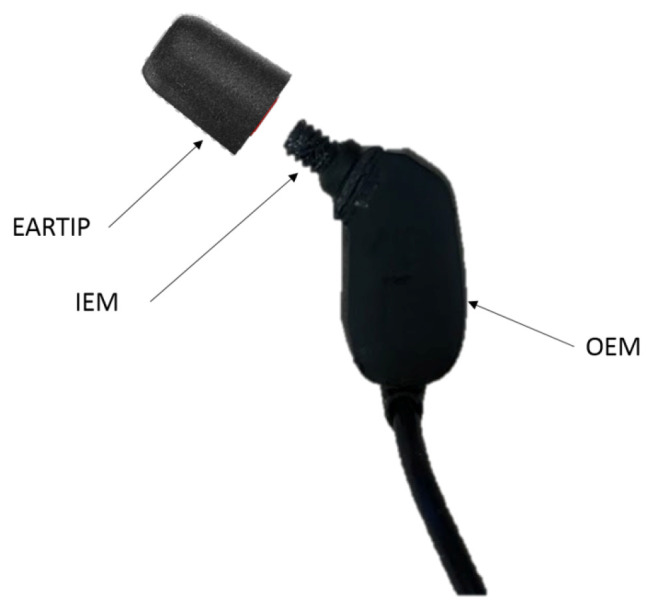
In-ear power sensor: Left cup earphone with foam tip.

**Figure 3 micromachines-11-01047-f003:**
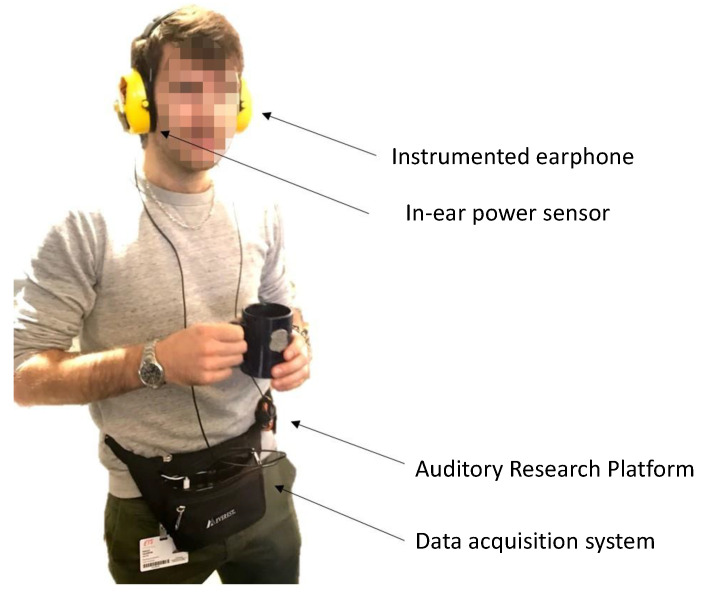
Test subject wearing the in-ear power sensor device and the Auditory Research Platform (ARP).

**Figure 4 micromachines-11-01047-f004:**
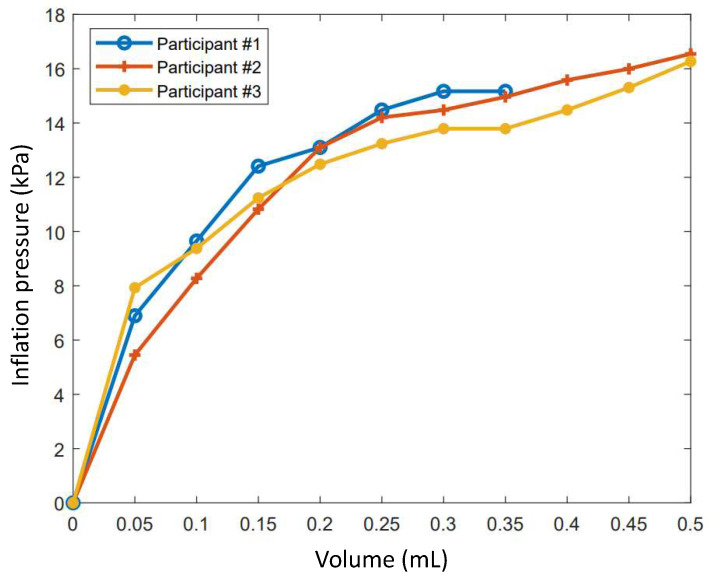
Earplug pressure–volume curves for three test subjects during inflation of the earplug within the earcanal.

**Figure 5 micromachines-11-01047-f005:**
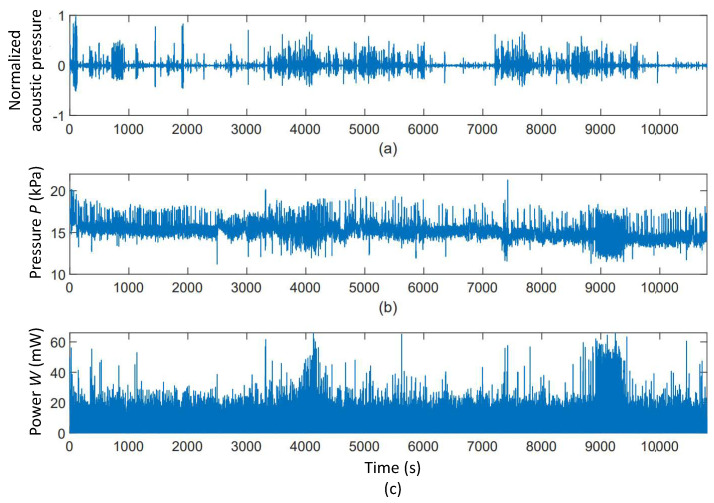
Audio, pressure and instantaneous power signals. (**a**) Audio, (**b**) Pressure and (**c**) Instantaneous power signals.

**Figure 6 micromachines-11-01047-f006:**
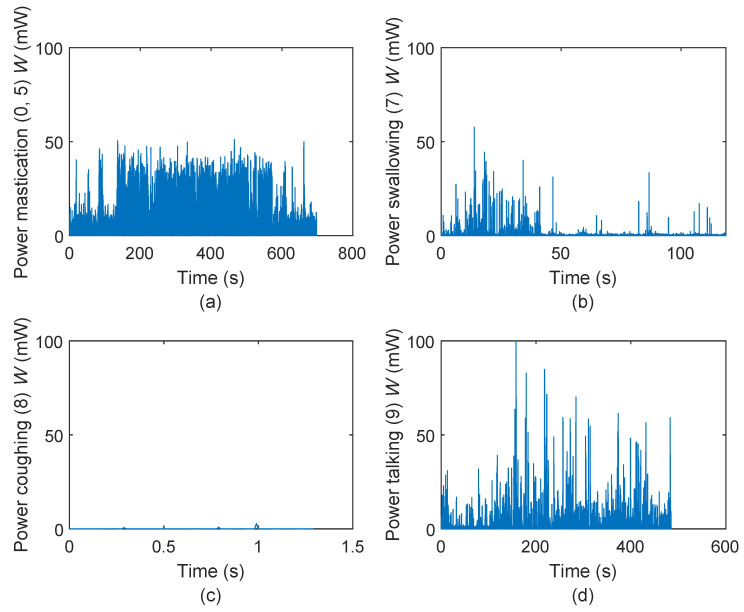
Instantaneous power associated with each jaw joint activity detected by the internal sound detection algorithm for one test subject: (**a**) chewing, (**b**) swallowing, (**c**) coughing, (**d**) talking.

**Figure 7 micromachines-11-01047-f007:**
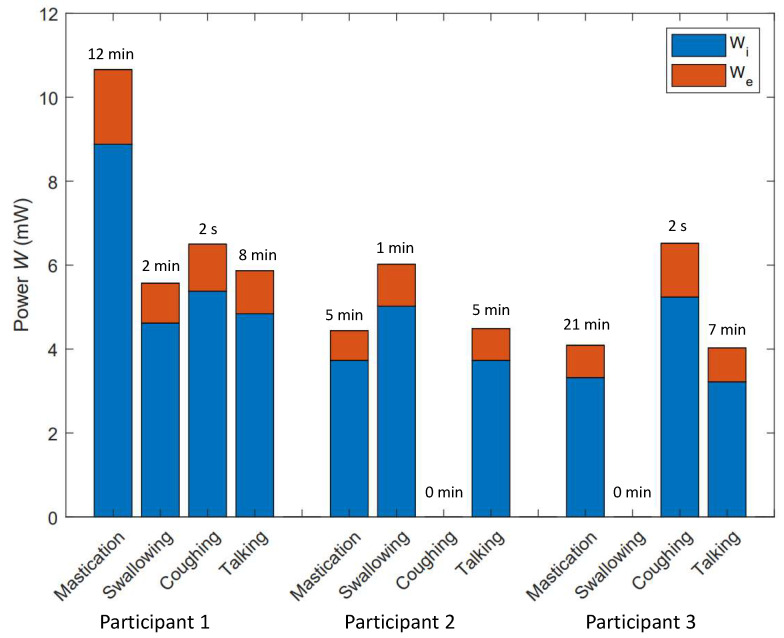
Mean power per jaw joint activity evaluated for three test subjects.

**Table 1 micromachines-11-01047-t001:** Verbal and nonverbal audio events detected by the classification algorithm.

Event	Id Number
Basic events:	
Clicking of teeth	0
Tongue clicking	1
Blinking forcefully	2
Closing the eyes	3
Closing the eyes forcefully	4
Grinding the teeth	5
Clearing the throat	6
Saliva noise	7
Coughing	8
Talking	9
Jaw joint activity events groups	
Mastication (eating)	0, 7
Swallowing	7
Coughing	8
Talking	9
No jaw joint activity	1, 2, 3, 4, 5, 6

**Table 2 micromachines-11-01047-t002:** Experimental results  in detail.

			Test Subject				
Parameter	Symbol	Unity	#1	#2	#3	Mean	SD
Water volume	*V*	mL	2.8	1.8	2.7	2.2	0.47
Equivalent rigidity	*k*	$\mathrm{GPa}/m^{3}$	44.1	51.3	36.4	43.9	7.4
Test duration	Δt	min	180	156	144	160	18
Energy	*E*	J	49.4	32.5	30.2	37.4	1.5
Mean power	$W_{\mathrm{mean}}$	mW	4.6	3.4	3.5	3.8	0.6
Time ratio	*R*	%	12	7	19	13	5

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
