# Peer review of "Mobile In-Ear Power Sensor for Jaw Joint Activity"

_micromachines, 2020, doi:10.3390/mi11121047_

Round 1

Reviewer 1 Report

The study shows that energy derived from the jaw movement can  generate power that can be derived to hearing aids. The study is well designed and despite the number of subjects being low (n=3) demonstrates that chewing can generate up to 26 mW. The study is not very original as this result has been shown previously. However the current study investigates other types of movement that could be associated to chewing. The limiting factor is here physiological. How long can energy be delivered? Although this is not of fundamental importance here, it will be  relevant when translated into a patient related environment. the conclusion could mention how authors anticipate answering this question

Author Response

Thank you for your comment. The authors totally agree with the reviewer that the human factors like ergonomics and comfort of the in-ear energy harvesting devices are important and should be thoroughly studied in the future. This point has been added to the conclusion of the paper.

Reviewer 2 Report

Very well done study, very important research in order to improve the acceptance of wearing hearing aids.

Is it possible to give an outlook to the transfer of generated power by jaw joint activity to the hearing aid?

Author Response

Thank you for your comment. The authors believe that the piezoelectric energy harvesting is an effective way to convert the kinetic energy of earcanal movements to electrical energy. This point has been added to the conclusion of the paper.